# STAT6 Signaling Mediates PPARγ Activation and Resolution of Acute Sterile Inflammation in Mice

**DOI:** 10.3390/cells10030501

**Published:** 2021-02-26

**Authors:** Ye-JI Lee, Bo-Min Kim, Young-Ho Ahn, Ji Ha Choi, Youn-Hee Choi, Jihee Lee Kang

**Affiliations:** 1Department of Physiology, College of Medicine, Ewha Womans University, Seoul 07804, Korea; shyzizibe@naver.com (Y.-J.L.); qhals96@ewhain.net (B.-M.K.); yc@ewha.ac.kr (Y.-H.C.); 2Inflammation-Cancer Microenvironment Research Center, College of Medicine, Ewha Womans University, Seoul 07804, Korea; yahn@ewha.ac.kr (Y.-H.A.); jihachoi@ewha.ac.kr (J.H.C.); 3Department of Molecular Medicine, College of Medicine, Ewha Womans University, Seoul 07804, Korea; 4Department of Pharmacology, College of Medicine, Ewha Womans University, Seoul 07804, Korea

**Keywords:** STAT6, PPARγ, efferocytosis, macrophages, resolution of inflammation

## Abstract

The signal transducer and activator of transcription 6 (STAT6) transcription factor promotes activation of the peroxisome proliferator-activated receptor gamma (PPARγ) pathway in macrophages. Little is known about the effect of proximal signal transduction leading to PPARγ activation for the resolution of acute inflammation. Here, we studied the role of STAT6 signaling in PPARγ activation and the resolution of acute sterile inflammation in a murine model of zymosan-induced peritonitis. First, we showed that STAT6 is aberrantly activated in peritoneal macrophages after zymosan injection. Utilizing *STAT6^−/−^* and wild-type (WT) mice, we found that STAT6 deficiency further enhanced zymosan-induced proinflammatory cytokines, such as tumor necrosis factor-α, interleukin (IL)-6, and macrophage inflammatory protein-2 in peritoneal lavage fluid (PLF) and serum, neutrophil numbers and total protein amount in PLF, but reduced proresolving molecules, such as IL-10 and hepatocyte growth factor, in PLF. The peritoneal macrophages and spleens of *STAT6^−/−^* mice exhibited lower mRNA and protein levels of PPARγ and its target molecules over the course of inflammation than those of WT mice. The deficiency of STAT6 was shown to impair efferocytosis by peritoneal macrophages. Taken together, these results suggest that enhanced STAT6 signaling results in PPARγ-mediated macrophage programming, contributing to increased efferocytosis and inflammation resolution.

## 1. Introduction

Peroxisome proliferator-activated receptor gamma (PPARγ) is a major regulator of fatty acid synthesis and storage and glucose metabolism [1]. Recently, the role of PPARγ in mediating responses to inflammation has gained interest. PPARγ is expressed on numerous immune cells, including monocytes/macrophages, platelets, T and B lymphocytes, and dendritic cells [2,3,4,5]. PPARγ exerts anti-inflammatory properties that can modulate the immune inflammatory response. PPARγ agonists have been shown to act as negative regulators of monocytes and macrophages and dose-dependently inhibit the production of proinflammatory cytokines, such as tumor necrosis factor (TNF)-α, interleukin (IL)-1β, and IL-6, in human monocytes [6,7]. Additionally, PPARγ and its ligands have been reported to participate in controlling an alternative M2 activation of monocytes and macrophages [8,9] and be important players in all stages of inflammation resolution. Furthermore, PPARγ activation contributes to increased efferocytic capability of macrophages via enhanced abundance of its target molecules, such as well-known efferocytic surface receptors CD36 and macrophage mannose receptor (MMR) [10,11,12]. The expression and activity of PPARγ in various cell types are strictly regulated [13,14,15], and a variety of endogenous ligands can activate PPARγ in different cell types [16,17,18]. However, PPARγ expression and the presence of appropriate ligands are usually insufficient to elicit optimal or maximal biological response [15].

Signal transducer and activator of transcription 6 (STAT6) is the signal mediator of IL-4 and IL-13 and promotes an anti-inflammatory process by inducing the development of Th2 lymphocytes and M2-type macrophages [19,20,21,22,23,24]. Szanto et al. [15] demonstrated that IL-4 signaling augments PPARγ activity in immune cells through an interaction between PPARγ and STAT6 on the promoters of PPARγ target genes, including *FABP4*. STAT6 interacts with PPARγ to elicit macrophage polarization toward an anti-inflammatory phenotype [25,26,27]. However, the relevance of the PPARγ-STAT6 interaction in disease pathogenesis, immunological responses, and therapeutic intervention has not been fully determined.

Here, we investigated the in vivo effect of the STAT6-PPARγ signaling axis to resolve acute sterile inflammation. First, we characterized STAT6 activation in peritoneal macrophage and spleen samples during zymosan-induced acute peritonitis. Next, applying *STAT6-*deficient *(STAT6*^-/-^) and wild-type (WT) mice, we examined the time course of the inflammatory response, PPARγ expression and activation, and the efferocytic ability of peritoneal macrophages. With these tools, we demonstrated that STAT6 signaling is required to restore PPARγ abundance and activation in peritoneal macrophages during acute sterile inflammation, leading to enhanced efferocytic ability and resolution of inflammation.

## 2. Materials and Methods

### 2.1. Reagents

Zymosan was purchased from Sigma-Aldrich (St. Louis, MO, USA). The antibodies used for Western blotting were as follows: anti-phospho STAT6 (Tyr-641), anti-STAT6, anti-PPARγ, anti-CD36, anti-MMR, and anti-arginase-1 (Arg1) from Cayman Chemical Co (Ann Arbor, MI, USA), anti-phospho JAK3 (Tyr-980/981), anti-JAK3 (Cell signalling Technology, Danvers, MA), and anti-β-actin from Sigma-Aldrich. Pierce BCA protein assay kit was purchased from Thermo Scientific (Rockford, IL, USA). The gene-specific relative RT-PCR kit was obtained from Invitrogen Life Technologies (Carlsbad, CA, USA). M-MLV reverse transcriptase was from Enzynomics (Seoul, Korea). DNA polymerase Klenow fragment and dNTPs were obtained from Intron Biotechnology (Seoul, Korea).

### 2.2. Animals

*STAT6^−/−^* and WT controls with identical background (BALB/c) mice were obtained from The Jackson Laboratory (Bar Harbor, ME, USA). Male *STAT6^−/−^* and WT mice aged 6–8 weeks were used for all experiments. The Animal Care Committee of the Ewha Medical Research Institute approved the experimental protocol. Mice were cared for and handled in accordance with the National Institutes of Health Guide for the Care and Use of Laboratory Animals.

### 2.3. Induction of Acute Sterile Inflammation and Treatment

*STAT6^−/−^* and WT mice were intraperitoneally (i.p.) administered 1 mg zymosan in 500 μL phosphate-buffered saline [28]. All animals were euthanized at 6, 24, or 72 h after zymosan injection.

### 2.4. Isolation of Peritoneal Lavage Cells, Spleen, and Lungs

The number of neutrophils and peritoneal macrophages in PLF were determined according to their unique cell diameter using an electronic Coulter counter fitted with a cell-sizing analyzer (Coulter Model ZBI with a channelizer 256; Beckman Coulter, Indianapolis, IN, USA). After peritoneal lavage, the spleen and lungs were removed, and immediately frozen in liquid nitrogen and stored at −70 °C.

### 2.5. Preparation of Peritoneal Macrophages

Peritoneal macrophages were cultured (5 × 10^5^ per well in 6-well plates) in serum-free X-VIVO medium for 60 min. Nonadherent cells were removed before isolation of total RNA and protein. Approximately 90–95% of the plastic-adherent cells were morphologically macrophages.

### 2.6. Measurement of Total Protein in Lavage Samples

Total protein concentration in lavage fluid was measured using the method devised by Hartree [29] with bovine serum albumin as a standard.

### 2.7. Enzyme-Linked Immunosorbent Assay (ELISA)

For serum cytokine quantification, blood was collected from mice via cardiac puncture, and the serum was separated by centrifugation at 1600× *g* for 5 min at 4 °C. The abundance of IL-4, IL-13, TNF-α, IL-6, macrophage inflammatory protein (MIP-2), IL-10, and hepatocyte growth factor (HGF) was quantitated in first cell-free PLF and the serum and by ELISA (R&D Systems, Minneapolis, MN, USA) following the manufacturer’s instructions.

### 2.8. Western Blotting

Spleen and lung tissue homogenates (10–50 μg protein/lane) were separated using an 8–10% SDS-PAGE gel and electrophoretically transferred onto nitrocellulose paper. The membranes were probed with specific antibodies to phospho-STAT6/SAT6, phospho-JAK3/JAK3, PPARγ, CD36, MMR, Arg1, or β-actin (1:1000 dilution) for 20 h, followed by the addition of secondary antibodies (1:1000) for 30 min. Detection was performed using an enhanced chemiluminescence detection kit (Thermo Scientific).

### 2.9. Real-Time Quantitative PCR

Gene expression was analyzed using real-time quantitative PCR (qPCR) on a StepOnePlus system (Applied Biosystems, Life Technologies, Carlsbad, CA, USA). For each qPCR assay, a total of 50 ng cDNA was used. The following primers were used (name: forward primer, reverse primer): (1) *PPAR**γ*: 5′-GCCCTTTGGTGACTTTATGG-3′, 5′-CAGCAGGTTGTCTTGGATGT-3′; (2) *CD36*: 5′- TTGTACCTATACTGTGGCTAAATGAGA-3′, 5′-CTTGTGTTTTGAACATTTCTGCTT-3′; (3) *MMR*: 5′-AGAAAATGCACAAGAGCAAGC-3′, 5′-GGAACATGTGTTCTGCGTTG-3′; (4) *Arg1*: 5′-GTGGGGAAAGCCAATGAAG-3′, 5′-GCTTCCAACTGCCAGACTGT-3′; (5) *IL-4*: 5′-CGAGCTCACTCTCTGTGGTG-3′, 5′-TGAACGAGGTCACAGGAGAA-3′; (6) *IL-13*: 5′- CCTGGCTCTTGCTTGCCTT-3′, 5′- GGTCTTGTGTGATGTTGCTCA-3′; (7) *HPRT*: 5′-CAGACTGAAGAGCTACTGTAATG-3′, 5′-CCAGTGTCAATTATATCTTCAAC-3′. mRNA levels were normalized on the basis of *HPRT* mRNA [30] and are reported as fold change in expression over the control group.

### 2.10. Immunocytochemistry

Immunocytochemistry was performed on samples obtained by peritoneal lavage cytocentrifugation followed by adherence of peritoneal macrophages to tissue culture dishes. The slides were then fixed with 4% paraformaldehyde, permeabilized with 0.1% Triton X-100 (Sigma-Aldrich), and stained with anti-macrophage-specific marker (Mac3; BD Pharmingen, San Jose, CA), mouse monoclonal anti-phospho-STAT6, anti-STAT6, or anti-PPARγ antibody overnight at 4 °C. Subsequently, cells were washed with phosphate-buffered saline three times and incubated with fluorescent isothiocyanate-conjugated donkey anti-rabbit IgG (Jackson ImmunoReseach, West Grove, PA, USA). The slides were mounted in Vectashield mounting medium with DAPI (Vector Laboratories, Inc., Youngstown, OH, USA). All slides were imaged using a confocal microscope (LSM5 PASCAL; Carl Zeiss, Jena, Germany) equipped with a filter set with excitation at 488 and 543 nm. The phosphor-STAT6, STAT6, and PPARγ stainings were quantified by creating masks and measuring the mean fluorescence intensity of each staining using the laser scanning microscopy image examiner software (Carl Zeiss).

### 2.11. Efferocytosis Assays

Peritoneal lavage cells were isolated and cytospun to assess phagocytic indices [31,32]. The phagocytic index was calculated using the following formula: [(number of apoptotic bodies)/(200 total macrophages)] × 100.

### 2.12. Statistical Analysis

Values are expressed as the means ± S.E.M. ANOVA was performed for comparisons of multiple parameters, and Tukey’s post-hoc test was applied where appropriate. Student’s t-test was used for comparisons of two sample means. A p-value less than 0.05 was considered statistically significant. All data were analyzed using JMP software (SAS Institute, Cary, NC, USA).

## 3. Results

### 3.1. STAT6 Is Activated in Peritoneal Macrophages after Zymosan Injection

To determine whether STAT6 activation is induced during acute sterile inflammation, we analyzed STAT6 phosphorylation in macrophages lavaged from peritonea after intraperitoneal injection of zymosan. Double immunofluorescence staining for phospho-STAT6 and Mac3 in peritoneal macrophages showed that phospho-STAT6 expression was observed in Mac3-positive macrophages from WT mice and enhances in a time-dependent manner and peaks 72 h after zymosan injection (Figure 1A). Phospho-STAT6 staining was shown to localize in the nuclei of these cells, indicating active STAT6. These data suggest that STAT6 activation in peritoneal macrophages is enhanced after in vivo treatment with zymosan at a relatively late time point.

STAT6 activation is initiated when IL-4 and IL-13 bind to their receptors, promoting an anti-inflammatory process by inducing the development of M2-type macrophages [19,20,21,22,23,24]. In this experimental model, the abundance of IL-4 and IL-13 mRNAs in peritoneal macrophages and the spleen (Figure 1B,C) and their protein levels in peritoneal lavage fluid (PLF) from zymosan-treated mice appear to be consistent with basal levels at each time point after zymosan treatment (Figure 1D). Thus, these data suggest that STAT6 activation is derived in an IL-4/IL-13-independent manner over the course of acute inflammation following zymosan injection.

### 3.2. STAT6 Deficiency Aggravates Acute Sterile Inflammation

To confirm whether STAT6 signaling is required for the optimal resolution of acute sterile inflammation, we examined the inflammatory responses after zymosan injection using *STAT6*^−/−^ and WT control mice. First, we confirmed STAT6 protein deficiency in peritoneal macrophages, spleens, and lungs of *STAT6*^−/−^ mice using immunocytochemistry or Western blot analysis (Figure 1E,F). In addition, we examined whether the kinase upstream of STAT6, JAK3 activation is enhanced following zymosan injection. Phosphorylation of JAK3 in spleen from both WT and *STAT6*^−/−^ mice was enhanced in a time-dependent manner following zymosan injection. These data suggest that JAK3 signaling mediates phosphorylation of STAT6 after zymosan injection (Figure 1G).

*STAT6*^−/−^ mice had higher production of pro-inflammatory cytokines, such as TNF-α, IL-6, and MIP-2 in PLF after zymosan injection than WT mice (Figure 2A–C). However, the enhanced amounts of IL-10 and HGF in PLF of WT mice after zymosan injection were reduced in that of *STAT6*^−/−^ mice (Figure 2D,E). Additionally, *STAT6*^−/−^ mice had higher neutrophil count and total protein content in PLF than WT mice (Figure 2F,G). Furthermore, enhanced proinflammatory cytokine production was observed in the serum of *STAT6*^−/−^ mice (Figure 2H–J). Taken together, these data indicate that inflammatory responses in PLF and serum are significantly aggravated in *STAT6*^−/−^ mice.

### 3.3. STAT6 Deficiency Leads to Decreased PPARγ Expression and Activation during Acute Sterile Inflammation

It remains poorly understood how PPARγ expression and activation are modulated by STAT6 in an in vivo acute inflammation model. We hypothesized that prolonged STAT6 activation would facilitate recovery and enhance PPARγ expression and activation during acute sterile inflammation. To test this hypothesis, we first assessed PPARγ abundance and functional activity in peritoneal macrophages and the spleen after zymosan injection. Immunocytochemistry analysis by confocal microscopy showed that the amount of PPARγ protein expression within the nuclei of peritoneal macrophages decreased at 6 h and then gradually increased as long as 72 h after zymosan injection (Figure 3A). Similar to the pattern of PPARγ protein expression, the mRNA levels of *PPARγ* in peritoneal macrophages were initially reduced at 6 h followed by a progressive time-dependent increase as long as 72 h after zymosan injection (Figure 3B). *STAT6*^−/−^ mice had significantly lower levels of PPARγ protein and mRNA expression in peritoneal macrophages than WT mice at each time point after zymosan injection (Figure 3A,B).

To investigate the functional activity of PPARγ, we examined the changes in the mRNA abundance of *CD36*, *MMR*, and *Arg1*, which are well-established direct transcriptional targets of PPARγ. Similarly, the mRNA amounts of *PPARγ*, *CD36*, *MMR*, and *Arg1* in peritoneal macrophages from *STAT6*^−/−^ mice were significantly less than those from WT mice during inflammation (Figure 3B). Additionally, a similar trend was observed in the spleen. Furthermore, the mRNA and protein levels of PPARγ, as well as these target genes, in the spleens of *STAT6*^−/−^ mice were lower than those in spleens of WT mice (Figure 3C,D). Notably, basal expression of PPARγ and its target genes at mRNA and protein levels was similar in peritoneal macrophages and spleen from *STAT6*^−/−^ mice compared with WT controls (Figure 3A–D). Taken together, the delayed acquisition of PPARγ expression and functional activity in *STAT6*^−/−^ mice is associated with exaggerated inflammatory cytokines and neutrophilic inflammation, indicating an association of the STAT6/PPARγ pathway to resolve acute sterile inflammation.

### 3.4. STAT6 Activation Is Associated with Enhanced Efferocytosis by Macrophages during Acute Sterile Inflammation

The ability of macrophages to clear apoptotic cells (efferocytosis) needed for resolving inflammation depends on their programming [26]. Using a genetic approach, we found that deficient STAT6 reversed the levels of efferocytic surface receptors and PPARγ target molecules, such as CD36 and MMR, during zymosan-induced inflammation. Thus, we examined the role of STAT6 in changing the efferocytic ability of peritoneal macrophages during zymosan-induced acute peritonitis. The efferocytic ability of macrophages lavaged from peritonea was examined microscopically for analysis of in vivo efferocytosis of endogenous apoptotic cells. Consistent with a previous report [28], detectable efferocytosis by macrophages in WT mice was progressively increased as long as 72 h after zymosan injection. Efferocytosis following zymosan injection was significantly lower in peritoneal macrophages from STAT6-deficient mice than in those from zymosan-only-treated or WT mice (Figure 4A,B).

## 4. Discussion

The goal of our study was to determine the role of STAT6 in PPARγ activation to resolve acute sterile inflammation in a zymosan-induced peritonitis murine model, which has been widely used as a self-resolving model of acute inflammation [33,34,35]. STAT6 activation is associated with tyrosine phosphorylation at specific sites, resulting in STAT6 dimerization and translocation into the nucleus to initiate transcription [36,37,38]. In the present study, we first demonstrated enhanced STAT6 phosphorylation (Y^641^) and nuclear translocation in peritoneal macrophages and spleens in a time-dependent manner after zymosan injection. Notably, neither the mRNA or protein levels of IL-4 and IL-13 increase in peritoneal macrophages, PLF, or the spleen following zymosan injection. These data suggest that IL-4/IL-13 production is not essential for STAT6 activation in zymosan-induced peritonitis. Several other cytokines, including IL-3/15, interferon alpha (IFN-α), and platelet-derived growth factor (PDGF), activate STAT6 in different cell types [39,40,41,42]. Among these cytokines, induction of PDGF and IFN-α by zymosan stimulation has been demonstrated in macrophages and dendritic cells, respectively [43,44]. In addition, recent data suggest that the membrane-associated proteins, including ATP-binding cassette transporter A1, thrombomodulin, and Annexin A1, are involved in STAT6/PPARγ signaling pathway [45,46,47,48]. Thus, further investigation is needed to determine whether and how these proteins participate in this pathway during zymosan-induced peritonitis.

Resolution of inflammation is a coordinated and active process, aimed at actively suppressing and extinguishing a vibrant inflammatory reaction [49]. Coordinated program of resolution rapidly initiates after acute challenges by cellular pathways. Cellular processes for onset and resolution of inflammation coexist during whole phases of inflammation. The process of resolution gradually enhanced while inflammatory reaction is gradually declined. In the present study, using a genetic approach, we explored whether enhanced STAT6 activation modulates inflammatory responses to zymosan injection. Furthermore, the role of STAT6 in resolving zymosan-induced acute inflammation was verified in experiments with STAT6-deficient mice. The production of proinflammatory cytokines, such as TNF-α, IL-6, and MIP-2, in PLF as well as serum from STAT6-deficient mice was greater than those from WT mice after zymosan injection. In contrast, the amounts of anti-inflammatory, pro-resolving molecules, including IL-10 and HGF, in the PLF from STAT6-deficient mice were lower than those from WT mice. Moreover, downregulatory effect of STAT6 deficiency on HGF proceeds up to 72 h after zymosan treatment. Similarly, Chen et al. [50] reported that further increases in proinflammatory cytokines, including IL-6 and monocyte chemoattractant protein-1, were observed in peritoneal macrophages from programmed cell death-1 knockout mice at 4 h after zymosan injection when levels of STAT6 phosphorylation in peritoneal macrophages were lower compared to those in WT mice. These data support the concept that downregulating phosphorylation of STAT6 leads to M1 polarization of macrophages.

Other key inflammatory components, including neutrophil numbers and total protein levels in PLF were further enhanced in STAT6-deficient mice compared with those in WT mice after zymosan injection. We demonstrated that STAT6 deficiency results in enhanced acute inflammatory responses following zymosan injection compared with WT controls. Acute inflammation is most often associated with a neutrophil-rich cellular infiltrate and is generally resolved in a period of days. Notably, the enhanced effects on neutrophil number, and protein levels prolonged until 72 h after zymosan treatment. In particular, the time interval for 50% reduction of the maximal recruitment of neutrophils was greater in *STAT6^−/−^* mice than in zymosan-only-treated and WT mice, suggesting a proresolving role of STAT6 [51,52]. Taken together, genetic evidence indicate that STAT6 signaling is required for the optimal resolution of zymosan-induced inflammation through the relative reduction of proinflammatory mediator production and increased anti-inflammatory mediator production, resulting in reduced neutrophil recruitment and shortened neutrophilia duration in zymosan-induced inflammation.

Data from in vitro studies using human macrophages and bone marrow-derived macrophages from *STAT6^−/−^* mice demonstrated the possibility that STAT6 augments PPARγ activity on target gene promoters [15]. Additionally, IL-4 augmented PPARγ response before inducing 15-lipoxygenase in human macrophages, suggesting that STAT6 is unlikely to act via ligand generation. These data also negated the hypothesis that STAT6 would induce degradation of a repressor for PPARγ or generate an activator for the retinoid X receptor, the permissive dimerization partner for STAT6. Importantly, Liao et al. [53] showed that transfection of STAT6 induced PPARγ promoter-driven luciferase activity in RAW264.7 cells, suggesting a possibility that STAT6 signaling enhances PPARγ expression via the inductive effect of STAT6 on the *PPAR**γ* promoter. STAT6 interacts with PPARγ to elicit macrophage polarization toward an anti-inflammatory phenotype [27]. Jun et al. [54] demonstrated that inhibition of STAT6 phosphorylation suppressed IL-4-activated PPARγ signaling pathways accordingly to increase lipid synthesis in human meibocytes. Thus, it is important to determine the relevance of STAT6-PPARγ interaction in disease pathogenesis, immunological responses, and therapeutic intervention.

Here, in a model of zymosan-induced peritonitis, we demonstrated the critical role of STAT6 activation in the recovery and enhancement of PPARγ expression and activation in peritoneal macrophages. The abundance of PPARγ mRNA and protein in peritoneal macrophages was significantly reduced at 6 h after zymosan injection and then progressively enhanced up to 72 h. The time course of the transcriptional induction of PPARγ targets, such as CD36, MMR, and Arg1, after zymosan injection paralleled these changes in PPARγ abundance, indicating a progressive enhancement of PPARγ activity due to its increased levels. Down-regulation of PPARγ expression has also been shown in hepatic residential macrophages Kupffer cells and RAW264.7 macrophages following in vitro stimulation with LPS and TNF-α [55,56]. Thus, TLR and cytokine-derived signaling in macrophages might be involved in the initial down-regulation of PPARγ expression and activity after zymosan treatment. Additionally, upregulation of PPARγ target gene expression indicates the characteristic of M2 macrophage programming over the course of acute inflammation after zymosan treatment [11,57,58]. However, the decreases in PPARγ expression and its target genes at mRNA or protein levels were confirmed in STAT6-deficient mice at each time point after zymosan injection. Given that PPARγ is known to inhibit activator protein 1, specificity protein-1, and nuclear factor-κB-driven proinflammatory cytokine transcription [11,59,60], it was hypothesized that elevated levels of mediators might reflect delayed or deficient acquisition of PPARγ expression and activity in STAT6^-/-^ mice. Fernandez-Boyanapalli et al. [28] showed that the levels of proinflammatory cytokines in PLF were significantly exaggerated in chronic granulomatous disease (CGD) mice at 6 h after zymosan injection and associated with delayed acquisition of PPARγ expression and activity in peritoneal macrophages. Taken together, these data provide physiological implications that restoration and progressive increases in PPARγ abundance over the course of acute inflammation are mediated by enhanced STAT6 signaling. While additional studies are clearly warranted, current observations raise the possibility that a cascade of inductive and cooperative interactions in the Stat6/PPARγ pathway may allow for optimal and sustained M2 activation. Notably, because PPARγ expression and functional activity were still observed in STAT6-deficient mice, non-STAT6-dependent signaling pathways are also involved in controlling PPARγ expression and activity in this experimental model.

Evidence suggests that plasticity in macrophage programming, in response to changing environmental cues, modulates efferocytic capability [26]. Activation of the nuclear receptors PPARγ, PPARδ, liver X receptor, and possibly retinoid X receptor alpha is essential to programming for enhanced efferocytosis [11,26,57,58,61]. The biological process of PPARγ activation likely involves ligand- or signaling-dependent transactivation of efferocytic surface receptors and markers alternative macrophage activation, such as CD36, MMR, and Arg1 [11,51,62]. Similarly, Majai et al. [11] demonstrated that the PPARγ antagonist during differentiation of human monocytes to macrophages significantly reduces efferocytic ability to engulf apoptotic neutrophils. Rocha et al. [48] reported that Annexin A1, a protein secreted by phagocytic cells, mediates efferocytosis in microglial cells via enhanced STAT6 phosphorylation along with increased PPARγ and CD36 expression. The increased PPARγ and CD36 expression appeared dependent on STAT6 phosphorylation. Thus, we propose that genetic deficiency of STAT6 influences the apoptotic cell clearance ability of peritoneal macrophages associated with the reduction of PPARγ activity and its target gene expression. Efferocytic ability was slightly decreased at 6 h after zymosan injection and thereafter enhanced in a time-dependent manner in WT macrophages, confirming a report by Fernandez-Boyanapalli et al. [28]. Moreover, we found that the efferocytic ability of peritoneal macrophages is impaired in STAT6-deficient mice, compared with zymosan-only-treated or WT mice, at each time point after zymosan injection. These data support that STAT6 signaling enhances the efferocytic ability through macrophage PPARγ-mediated programming during acute inflammation and consequently, resolving inflammation. Notably, the continuous enhanced PPARγ expression and activation may occur after an initial decrease through a feedback loop by enhanced apoptotic cell clearance during inflammation via cross-talk between apoptotic cells and STAT6 signaling [10].

## 5. Conclusions

In summary, we demonstrate that enhanced STAT6 activation during acute sterile inflammation plays an essential role in resolving acute sterile inflammation via recovery and enhancement of PPARγ expression and the activation and enhanced efferocytosis of macrophages. Nonetheless, direct and indirect mechanisms for STAT6-mediated PPARγ facilitation in macrophages should be further studied in the context of in vitro and in vivo treatment with zymosan. Future studies investigating the enhanced STAT6/PPARγ signaling may contribute to the development of potential candidates in therapeutic strategies to restore tissue homeostasis after an inflammatory event.

## Figures and Tables

**Figure 1 cells-10-00501-f001:**
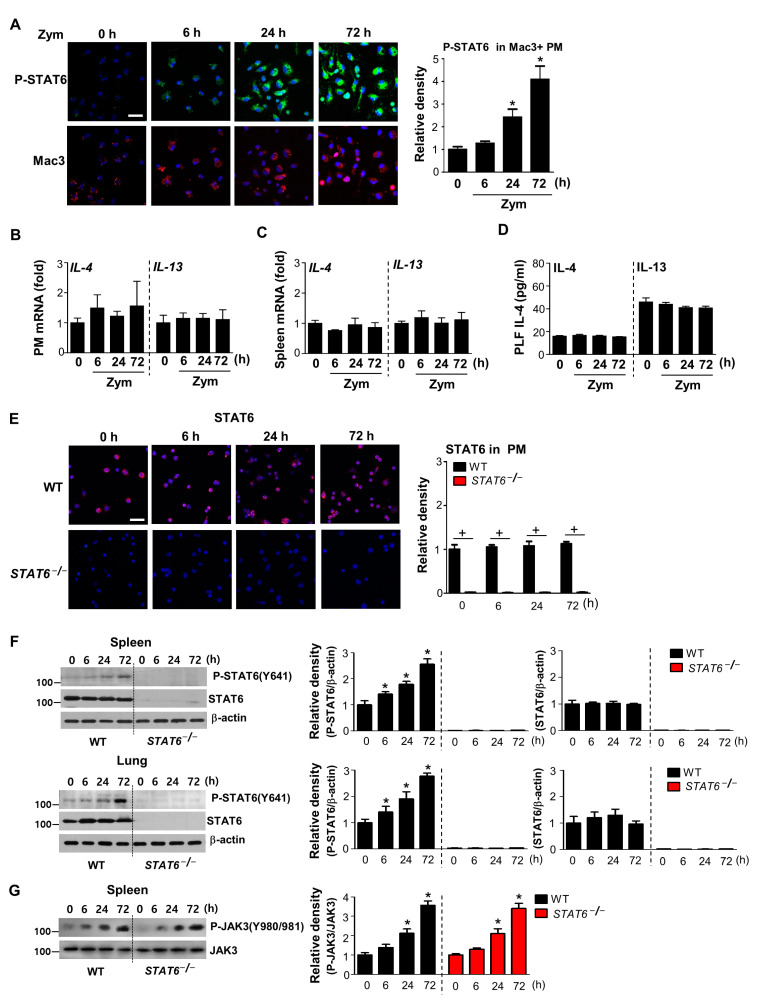
Enhanced STAT6 activation in peritoneal macrophages from WT mice and STAT6 deficiency in peritoneal macrophages, spleen, and lungs from *STAT^−/−^* mice during zymosan-induced peritonitis. Wild-type (WT) and *STAT6^−/−^* mice were injected i.p. with 1 mg zymosan (Zym) in 500 μL saline, and peritoneal lavage supernatant, spleens and lungs were collected at 6, 24, or 72 h after Zym injection. (**A**) Left: Immunofluorescence staining for phospho-STAT6 (green) and macrophage-specific marker (Mac3, Red) in peritoneal macrophages (PM) from WT mice. Images were captured at 400× magnification. Right: Quantification of phospho-STAT6 staining in Mac3-positive macrophages. The imaging medium was Vectashield fluorescence mounting medium containing DAPI. Scale bars = 50 μm. Representative results from three mice per group are shown. (**B**,**C**) The levels of *IL-4* and *IL-13* mRNA over time in PM and spleens analyzed by real-time PCR and normalized to that of hypoxanthine guanine phosphoribosyl transferase (Hprt) mRNA. (**D**) The abundance of IL-4 and IL-13 in peritoneal lavage fluid (PLF) as assessed by ELISA. (**E**) Left: Immunofluorescence staining for STAT6 (green) in peritoneal macrophages; 400× magnification. Right: Quantification of STAT6 staining using Vectashield fluorescence mounting medium containing DAPI. Scale bars = 50 μm. Representative results from three mice per group are shown. (**F**) Left: Western blot analysis of total and phosphorylated STAT6 in spleen and lung homogenates. Right: Densitometric analysis of the relative abundance of total and phosphorylated STAT6 normalized to that of β-actin at the indicated times. (**G**) Left: Western blot analysis of total and phosphorylated JAK3 in spleen. Right: Densitometric analysis of the relative abundance of phosphorylated JAK3 normalized to that of total JAK3 at the indicated times. Values represent the means ± S.E.M. of three (**A**,**E**–**G**) or five mice (**B**–**D**) per group. * *p* < 0.05 compared with saline control; + *p* < 0.05 for *STAT*^−/−^ mice vs. WT mice at a given time point.

**Figure 2 cells-10-00501-f002:**
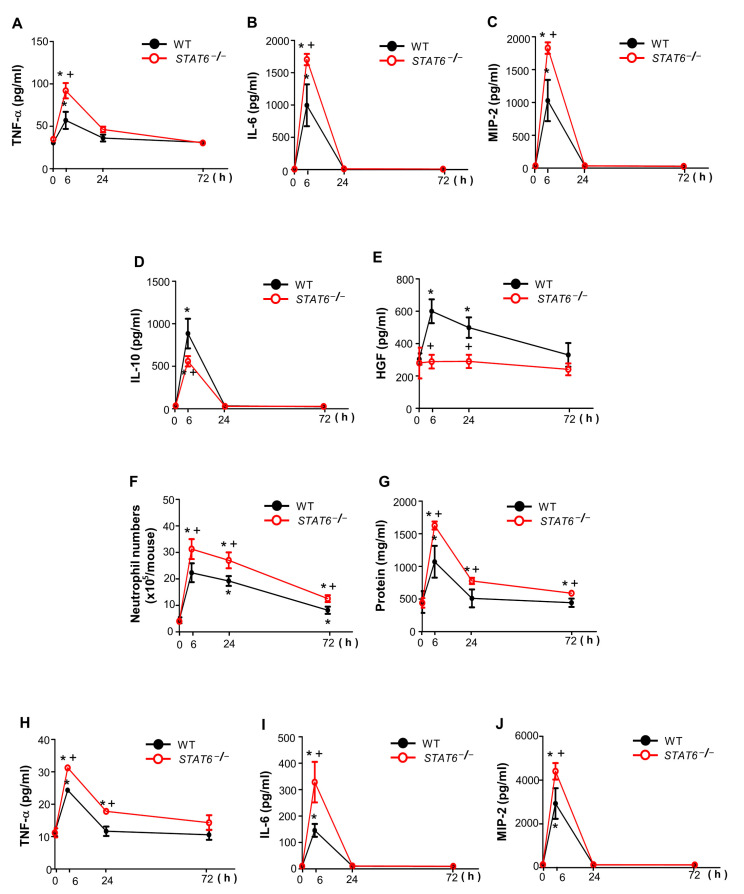
The inflammatory response was exacerbated in *STAT6^−/−^* mice. Peritonitis was induced as in Figure 1. The abundance of TNF-α (**A**), IL-6 (**B**), MIP-2 (**C**), IL-10 (**D**), and HGF (**E**) in peritoneal lavage fluid (PLF) as assessed by ELISA. Neutrophil count (**F**) and total protein abundance in PLF (**G**). The abundance of TNF-α (**H**), IL-6 (**I**), and MIP-2 (**J**) in serum as assessed by ELISA. Values represent the means ± S.E.M. of five mice per group. * *p* < 0.05 compared with saline control; + *p* < 0.05 for *STAT^−/−^* mice vs. WT mice at a given time point.

**Figure 3 cells-10-00501-f003:**
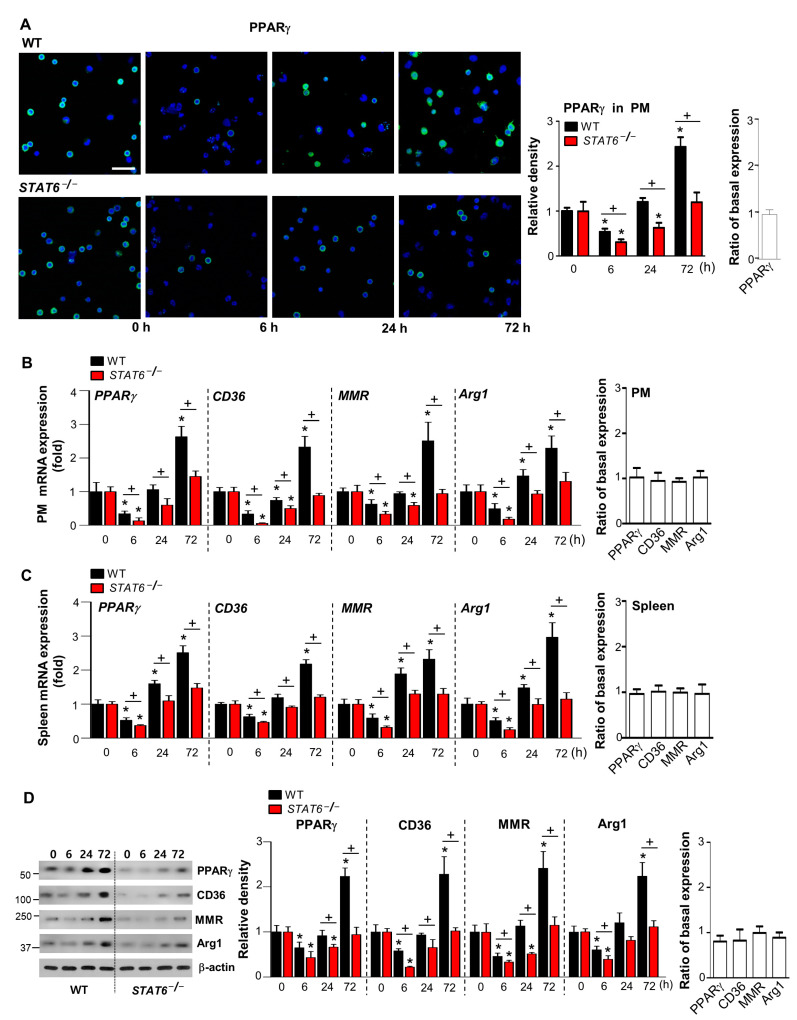
PPARγ expression and activation are decreased in peritoneal macrophages from *STAT6^-/-^* mice. Peritonitis was induced as in Figure 1. (**A**) Left: The mean fluorescence intensity (MFI) ratio of immunofluorescence staining for PPARγ (green) and DAPI (blue) for the nuclei in peritoneal macrophages (PM) from mice injected with zymosan at the indicated times; 400× magnification. Scale bars = 50 μm. Results are representative of three mice at each time point after zymosan treatment. Middle: Quantification of PPARγ staining. Right: The mean fluorescence intensity (MFI) ratio of basal PPARγ protein expression in PM from *STAT6*^−/−^ mice compared with WT controls. (**B**) Left: Changes in the levels of *PPARγ*, *CD36*, *MMR*, and *Arg1* mRNAs over time in PM were analyzed by real-time PCR and normalized to that of *Hprt* mRNA. Right: The ratio of basal mRNA expression in PM from *STAT6*^−/−^ mice compared with WT controls. (**C**) Left: Changes in the levels of *PPARγ*, *CD36*, *MMR*, and *Arg1* mRNAs over time in spleens were analyzed by real-time PCR and normalized to that of *Hprt* mRNA. Right: The ratio of basal mRNA expression in spleens from *STAT6*^−/−^ mice compared with WT controls. (**D**) Left: Western blot analysis of PPARγ, CD36, MMR, and Arg1 in spleen homogenates. Middle: Densitometric analysis of the relative abundance of the indicated proteins normalized to that of β-actin. Right: The ratio of basal protein expression in spleen homogenates from *STAT6*^−/−^ mice compared with WT controls. The values represent the means ± S.E.M. of at least three (**A**,**D**) or five mice in each group (**B**,**C**). * *p* < 0.05 compared with saline control; + *p* < 0.05 for *STAT6^−/−^* mice vs. WT mice at a given time point.

**Figure 4 cells-10-00501-f004:**
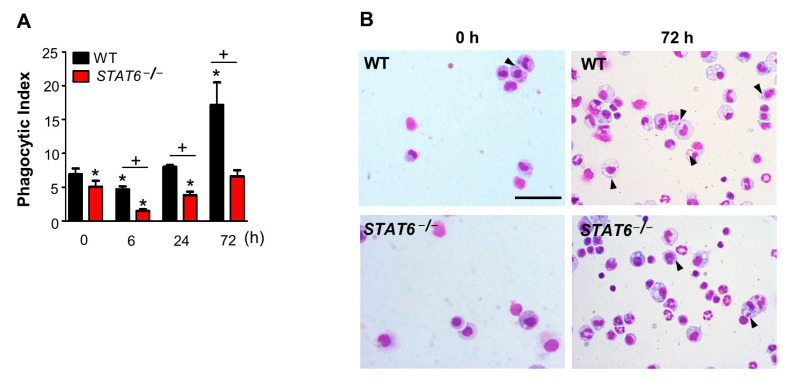
STAT6 reduces efferocytic ability of peritoneal macrophages during zymosan-induced acute inflammation. Peritonitis was induced as in Figure 1. (**A**) Phagocytic indices in peritoneal lavaged macrophages. (**B**) Representative photomicrographs (400× magnification) showing cytospun-stained peritoneal lavaged cells at 72 h after zymosan injection. Arrowheads indicate peritoneal macrophages with engulfed apoptotic cells or fragments. Scale bars = 200 μm. Values represent the means ± S.E.M. of five mice per group. * *p* < 0.05 compared with saline control; + *p* < 0.05 for *STAT6^−/−^* mice vs. WT mice at a given time point.

## Data Availability

All data presented within this study are available within the manuscript.

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
