# Peer review of "STAT6 Signaling Mediates PPARγ Activation and Resolution of Acute Sterile Inflammation in Mice"

_cells, 2021, doi:10.3390/cells10030501_

Round 1
Reviewer 1 Report
The goal of the study by Lee et al was to examine the role of STAT6 signaling in PPARγ activation and the resolution of acute sterile inflammation in a murine model of zymosan‐induced peritonitis. The authors bring an interesting connection between STAT6 and PPARg. This becomes even more interesting when STAT6 activation may be independent of IL-4. Indeed, STAT6 activation has always been associated with IL-4/IL-4R signaling and thus this study appears to bring a new twist to the mechanism by which STAT6 is activated. However, the quality of the data and the rationale for the study is rather weak and the conclusions are not supported by the provided data. The following are some (but not all) of the concerns:
Major concerns:
- The data in Figure 1, especially 1A and B, is problematic. There is no evidence that the collected cells are macrophages. A close examination of the cells appears to indicate that most of the cells with (+) p-STAT6 signal are PMNs. Also, at 72 h, the phosphorylation in mononucleated cells is primarily cytoplasmic. More evidence demonstrating that what is presented is actually p-STAT6 and that it is indeed in macrophages is needed.
- The levels of IL-4 and 13 in PLF is excessively high in control mice; it may be because of the mouse background (although this reviewer did check the literature and the numbers do not seem to be that high). This also becomes problematic because the mice are already skewed toward Th2 inflammation. The use of C57BL/6 mice may be more appropriate.
- It seems that the cytospin/cells at 72 h have a problem. Better samples and images are required
- Data in Figure 2 is predictable and brings no new information. Stat6 phosphorylation is expected to be absent in Stat6 KO cells. The authors should consider combining Fig 1 with Fig 2.
- The data in Figure 4 is also problematic. The authors do not differentiate between induction of inflammation and its resolution. There is an aggravation of inflammation; however, the resolution (~72 h) in WT and STAT6 ko upon treatment with zymosan is very similar between the two strains.
- The author should address some of the key publications that make the connection between STAT6 and PPARg as well as STAT6 and the zymosan model. This includes the study by da Rocha et al. [Cell Biochem Funct. 2019 Oct;37(7):560-568. Control of expression and activity of peroxisome proliferated-activated receptor γ by Annexin A1 on microglia during efferocytosis]. Here an association between anexin and STAt6 activation and PPARg was established in efferocytosis. The authors should also address the study by Rao et al (J Immunol 2019 Aug 15;203(4):1031-1043) in which a connection with STAT6 and resolution of zymosan-induced inflammation was examined.
Author Response
The goal of the study by Lee et al was to examine the role of STAT6 signaling in PPARγ activation and the resolution of acute sterile inflammation in a murine model of zymosan‐induced peritonitis. The authors bring an interesting connection between STAT6 and PPARg. This becomes even more interesting when STAT6 activation may be independent of IL-4. Indeed, STAT6 activation has always been associated with IL-4/IL-4R signaling and thus this study appears to bring a new twist to the mechanism by which STAT6 is activated. However, the quality of the data and the rationale for the study is rather weak and the conclusions are not supported by the provided data. The following are some (but not all) of the concerns:
Major concerns:
(C1) The data in Figure 1, especially 1A and B, is problematic. There is no evidence that the collected cells are macrophages. A close examination of the cells appears to indicate that most of the cells with (+) p-STAT6 signal are PMNs. Also, at 72 h, the phosphorylation in mononucleated cells is primarily cytoplasmic. More evidence demonstrating that what is presented is actually p-STAT6 and that it is indeed in macrophages is needed.
(R1) We appreciate the reviewer’s suggestion. As suggested, to give more evidence demonstrating p-STAT6 in macrophages, we have performed double immunofluorescence staining for phospho-STAT6 and macrophage-specific marker (Mac3) in peritoneal macrophages. We have described added the results as follows (lines 159-162): “Double immunofluorescence staining for phospho-STAT6 and Mac3 in peritoneal macrophages showed that phospho-STAT6 expression was observed in Mac3-positive macrophages from WT mice enhances in a time-dependent manner and peaks 72 h after zymosan injection (Fig. 1A).
(C2) The levels of IL-4 and 13 in PLF is excessively high in control mice; it may be because of the mouse background (although this reviewer did check the literature and the numbers do not seem to be that high). This also becomes problematic because the mice are already skewed toward Th2 inflammation. The use of C57BL/6 mice may be more appropriate.
(R2) We appreciate the reviewer’s suggestion. Diverse mouse strains, including BALB/c, C57BL/6, Swiss-Webster, and C3 mice have been used for sterile or non-sterile inflammatory models. As suggested, we found that the levels of IL-4 and IL-13 in PLF from BALB/c which we used in the present studies were a little higher than those control C57BL/6 mice which were used by Walker et al. (J Immunol 1998, 161:1962 ) and Shang et al. (Front Endo 2019, 10:36) [16 vs. 11 pg/ml for IL-4; 46 vs. ~20 pg/ml for IL-13]. However, the protein levels of IL-4 were similar and IL-13 were lower in bronchoalveolar lavage fluid from control BALB/c mice compared to those from control C57BL/6 mice, respectively, [~100 pg/ml for IL-4; 33 vs. 120 pg/ml for IL-13] (Jin et al., Mol Med 2019, 19: 895; Wu et al., Life Sci 2019, 239:117067). Based on these data, it doesn’t seem likely that the differences of basal levels between such mouse strains are problematic.
(C3) It seems that the cytospin/cells at 72 h have a problem. Better samples and images are required
(R3) As suggested, we replaced the cytospin/cell at 72 h in Fig. 1A and Fig. 4B.
(C4) Data in Figure 2 is predictable and brings no new information. Stat6 phosphorylation is expected to be absent in Stat6 KO cells. The authors should consider combining Fig 1 with Fig 2.
(R4) As suggested, we combined Fig.1 with Fig. 2.
(C5) The data in Figure 4 is also problematic. The authors do not differentiate between induction of inflammation and its resolution. There is an aggravation of inflammation; however, the resolution (~72 h) in WT and STAT6 ko upon treatment with zymosan is very similar between the two strains.
(R5) We appreciate the reviewer’s suggestion. Resolution of inflammation is a coordinated and active process, aimed at actively suppressing and extinguishing a vibrant inflammatory reaction (Ortega-Gomez et al., EMBO Mol Med 2013; 5: 661). Coordinated program of resolution rapidly initiates after acute challenges by cellular pathways. Cellular processes for onset and resolution of inflammation coexist during whole phases of inflammation. The process of resolution gradually enhanced while inflammatory reaction is gradually declined. Our present study demonstrated that STAT6 deficiency upregulates proinflammatory mediators and neutrophil number but downregulates anti-inflammatory mediators in peritoneal lavage fluid. Notably, the enhanced effects on neutrophil number, and protein levels prolonged until 72 h after zymosan treatment. In particular, STAT6 deficient mice showed that increase in the time interval from the maximum neutrophil point to the 50% reduction point, suggesting a pro-resolution role of STAT6 (Serhan et al., Nature 2008, 8:349). Moreover, downregulatory effect of STAT6 deficiency on efferocytic ability of macrophages and an anti-inflammatory proresolving molecule, HGF, proceeds up to 72 h after zymosan treatment. Collectively, these data suggest that STAT6 signaling is required for the optimal resolution of zymosan-induced inflammation.
(C6) The author should address some of the key publications that make the connection between STAT6 and PPARg as well as STAT6 and the zymosan model. This includes the study by da Rocha et al. [Cell Biochem Funct. 2019 Oct;37(7):560-568. Control of expression and activity of peroxisome proliferated-activated receptor γ by Annexin A1 on microglia during efferocytosis]. Here an association between anexin and STAt6 activation and PPARg was established in efferocytosis. The authors should also address the study by Rao et al (J Immunol 2019 Aug 15;203(4):1031-1043) in which a connection with STAT6 and resolution of zymosan-induced inflammation was examined.
(R6) According to the Reviewer’s suggestion, we have addressed findings from Rocha et al. in the Discussion (lines 380-384). Actually, data from Rao et al. suggested that JAK3/STAT6 signaling mediates 15-LOX-1 expression and involvement of specialized proresolving mediators including lipoxins, resolvins, maresins, and protectins. They also demonstrated a role of V-ATPase for resolution of inflammation in zymisan-induced inflammation via the induction of 15-LOX-1 but not STAT6 signaling. Thus, we have added this publication at appropriate position in the Introduction of our revised manuscript.
We very much appreciate the suggestions raised by the reviewer. We hope that our manuscript will now be accepted for publication in Cells.
Sincerely,
Jihee Lee Kang, MD, PhD
Corresponding Author
Professor
Department of Physiology
College of Medicine
Ewha Womans University
Reviewer 2 Report
Lee and coauthors show a proresolving role for STAT6 in a mouse model of acute inflammation, a role linked to STAT6 phosphorylation and PPARg activity. Overall, the manuscript is written well, the experiments are nicely executed, and the data are clearly presented. In a few areas related to the mechanistic connection between PPARg and STAT6, more evidence or literature discussion would benefit the manuscript.
Comments:
Please clarify/elaborate more on the regulation of PPARg basal expression and activity in STAT6-/- mice. Where appropriate in Fig. 4, please show the difference in basal levels between WT and STAT6-/- by normalizing to WT 0 (h).
Although STAT6 is required for basal PPARg expression, the association between PPARg and pSTAT6 is less clear. Is there evidence for a causative link between STAT6 phosphorylation and PPARg activity? Are pSTAT6 or PPARg responsive to cytokine stimulation in cultured, zymosan-naive peritoneal macrophages? Please either provide experimental evidence or further clarify and expand on these subjects in the discussion.
Author Response
Lee and coauthors show a proresolving role for STAT6 in a mouse model of acute inflammation, a role linked to STAT6 phosphorylation and PPARg activity. Overall, the manuscript is written well, the experiments are nicely executed, and the data are clearly presented. In a few areas related to the mechanistic connection between PPARg and STAT6, more evidence or literature discussion would benefit the manuscript.
Comments:
(C1) Please clarify/elaborate more on the regulation of PPARg basal expression and activity in STAT6-/- mice. Where appropriate in Fig. 4, please show the difference in basal levels between WT and STAT6-/- by normalizing to WT 0 (h).
(R1) We appreciate the reviewer’s suggestion. As suggested, we clarified more on the regulation of PPARg basal expression and activity in STAT6-/- mice. We have added this issued in the Results as follows: “Notably, basal expression of PPARg and its target genes at mRNA and protein levels was similar in peritoneal macrophages and spleen from STAT6−/− mice compared with WT controls (Fig. 3A-D).”
(C2) Although STAT6 is required for basal PPARg expression, the association between PPARg and pSTAT6 is less clear. Is there evidence for a causative link between STAT6 phosphorylation and PPARg activity? Are pSTAT6 or PPARg responsive to cytokine stimulation in cultured, zymosan-naive peritoneal macrophages? Please either provide experimental evidence or further clarify and expand on these subjects in the discussion
(R2-1) We appreciate the reviewer’s suggestion. As we demonstrated in the Introduction, IL-4 signaling augments PPARg activity in immune cells through an interaction between PPARg and STAT6 on the promoters of PPARg target genes, including FABP4 (Szanto et al. Immunity 2010, 33:699). Thus, STAT6 acts as a facilitating factor for PPARg by promoting DNA binding and consequently increasing the number of regulated genes and the magnitude of responses. Furthermore, Liao et al. (J Clin Invest 2011, 121:2736) have shown that transfection of STAT6 induced PPARg promoter-driven luciferase activity in RAW264.7 cells, suggesting a possibility that STAT6 signaling enhances PPARg expression via the inductive effect of STAT6 on the PPARg promoter. STAT6 interacts with PPARg to elicit macrophage polarization towards an anti-inflammatory phenotype (Sajic et al., Sci Report 2013, 3:2350). Previous our findings showed that pharmacologic inhibition of STAT6 phosphorylation reduced enhanced PPARg expression and activation in macrophages by in vivo treatment with apoptotic cells, resulting in delaying the resolution of bleomycin-induced lung inflammation and fibrosis (Kim et al., Cell Physiol Biochem 2018, 45:1863). Jun et al. demonstrated that inhibition of STAT6 phosphorylation suppressed IL-4-activated the PPARg signaling pathways accordingly to increase lipid synthesis in human meibocytes (Ocul Surf 2020, 18:575). Rocha et al. reported that ANXA1, a protein secreted by phagocytic cells, mediates efferocytosis in microglial cells via enhanced STAT6 phosphorylation along with increased PPARg and CD36 expression. The increased PPARg and CD36 expression appeared dependent on STAT6 phosphorylation (Cell Biochem Funct 2019, 37:560). While additional studies are clearly warranted, current observations raise the possibility that a cascade of inductive and cooperative interactions in the Stat6/PPARg pathway may allow for optimal and sustained M2 activation. To support evidence for a causative link between STAT6 phosphorylation and PPARg activity, we have added these findings at appropriate positions in the Discussion.
(R2-2) Chen et al. reported that IL-6 and MCP-1 were increased quickly in peritoneal macrophages at 4 h after zymosan injection and then decreased at 48 and 72 h (Cell Death Dis 2016). Interestingly, further increases in these cytokine levels were observed in peritoneal macrophages from programmed cell death-1 (PD-1) KO mice at 4 h after zymosan injection when levels of STAT6 phosphorylation in peritoneal macrophages were lower compared those in WT mice. These data support the concept that downregulating phosphorylation of STAT6 leads to M1 polarization of macrophages. As discussed in the Discussion of our manuscript, Fernandez-Boyanapalli et al. (Blood 2010) showed that the levels of proinflammatory cytokines in PLF were significantly exaggerated in chronic granulomatous disease (CGD) mice after zymosan injection and associated with delayed acquisition of PPARg expression and activity in peritoneal macrophages. However, they did not measure these cytokine levels in macrophage cultured media. We have stated further this subject into the Discussion.
We very much appreciate the suggestions raised by the reviewer. We hope that our manuscript will now be accepted for publication in Cells.
Sincerely,
Jihee Lee Kang, MD, PhD
Corresponding Author
Professor
Department of Physiology College of Medicine
Ewha Womans University
Reviewer 3 Report
In this manuscript, the authors examined the role of STAT6 in acute inflammation models using STAT6-/- mice. Zymosan treatment induced STAT6 phosphorylation in peritoneal macrophages without induction of IL-4 or IL-13 expression. Proinflammatory cytokine levels were higher and levels of IL-10 and HGF were lower in STAT6-/- mice compared to WT mice. In peritoneal macrophages of STAT6-/- mice, expression of PPARg and its target genes was decreased and efferocytosis was also impaired. The authors suggest that STAT6 signaling is involved in PPARg-mediated macrophage function, such as efferocytosis and inflammation resolution.
- Fig. 1. What mechanism activates STAT6 signaling after zymosan treatment? What kinase(s) induce phosphorylation of STAT6?
- Fig. 3. Changes of cytokine production were observed 6 hours after zymosan treatment. At this time point, PPARg expression was decreased in both WT and STAT6-/- cells. These results indicate that the effect of cytokine production by STAT6 signaling is NOT mediated by PPARg (?). Please show direct evidence showing the involvement of PPARg in cytokine expression.
- STAT6 phosphorylation was increased 24 hours after zymosan treatment (Fig. 2). How do the authors explain the involvement of STAT6 signaling in regulation of cytokine production in Fig. 3?
- Fig. 4. The authors suggest that STAT6 is involved in PPARg transcriptional activity. However, mRNA expression of PPARg was decreased in STAT6-/- cells. Is PPARg a STAT6 target gene? Are there STAT-binding element(s) in the PPARg promoter?
- Fig. 5. How about the effect of PPARg ligand, such as pioglitazone, on efferocytic ability of peritoneal macrophages from WT and STAT6-/- mice stimulated with zymosan.
- There are clear phenotypes of STAT6-/- cells and mice in Figs. 2-5, indicating the involvement of STAT6 signaling. However, there is no evidence showing that STAT6 signaling is involved in PPARg-mediated macrophage function (cytokine production and efferocytosis). Please make the causal relationship clearer.
Author Response
In this manuscript, the authors examined the role of STAT6 in acute inflammation models using STAT6-/- mice. Zymosan treatment induced STAT6 phosphorylation in peritoneal macrophages without induction of IL-4 or IL-13 expression. Proinflammatory cytokine levels were higher and levels of IL-10 and HGF were lower in STAT6-/- mice compared to WT mice. In peritoneal macrophages of STAT6-/- mice, expression of PPARg and its target genes was decreased and efferocytosis was also impaired. The authors suggest that STAT6 signaling is involved in PPARg-mediated macrophage function, such as efferocytosis and inflammation resolution.
(C1) Fig. 1. What mechanism activates STAT6 signaling after zymosan treatment? What kinase(s) induce phosphorylation of STAT6?
(R1-1) In this study, we did not elucidate mechanism activating STAT6 signaling after zymosan treatment. As demonstrated, amounts of IL-4 and IL-13 at protein and mRNA levels in peritoneal macrophages and spleen, respectively, were not significantly changed following zymosan treatment. We have added this issue in the Discussion as follows (lines 295-300): “Among these cytokines, induction of PDGF and IFN-α by zymosan stimulation has been demonstrated in macrophages and dendritic cells, respectively [43, 44]. In addition, recent data suggested that the membrane-associated proteins, including ATP-binding cassette transporter A1 (ABCA1), thrombomodulin, and Annexin A1, are involved in STAT6/PPARg signaling pathway [45-48]. Thus, further investigation is needed to determine whether and how these proteins participate in this pathway during zymosan-induced peritonitis.”
(R1-2) In addition, we examined whether the kinase upstream of STAT6, JAK3 activation is enhanced following zymosan injection. Phosphorylation of JAK3 in spleen from both WT and STAT6−/− mice was enhanced in a time-dependent manner following zymosan injection. These data suggest that JAK3 signaling mediates phosphorylation of STAT6 after zymosan injection (Fig. 1G).
(C2) Fig. 3. Changes of cytokine production were observed 6 hours after zymosan treatment. At this time point, PPARg expression was decreased in both WT and STAT6-/- cells. These results indicate that the effect of cytokine production by STAT6 signaling is NOT mediated by PPARg (?). Please show direct evidence showing the involvement of PPARg in cytokine expression.
(R2) As commented that activation of STAT6 is increased slightly and PPARg expression and activity were reduced at 6 h after zymosan treatment. However, significant inhibition of PPARg expression and activation in peritoneal macrophages as well as spleen was shown at 6 h after zymosan treatment in STAT6-/- mice compared with those in wild-type mice. These data suggest that STAT6 signaling is required for the recovery of PPARg expression and activity after zymosan stimulation in vivo. Our present study also demonstrated that STAT6 deficiency upregulates proinflammatory mediators and neutrophil number but downregulates anti-inflammatory mediators in peritoneal lavage fluid. We further stated regarding this issue in The Discussion as follows (lines 356-363): “Given that PPARg is known to inhibit activator protein 1, specificity protein-1, and nuclear factor-κB driven proinflammatory cytokine transcription [11, 57, 58], it was hypothesized that elevated levels of mediators might reflect delayed or deficient acquisition of PPARg expression and activity in STAT6-/- mice.”
Down-regulation of PPARg expression has also been shown in hepatic residential macrophages Kupffer cells (KCs) and RAW264.7 macrophages following in vitro stimulation with LPS and TNF-a (Jhou et al., Am J Physiol Regul Integr Comp Physiol 2007; Miksa et al., J Immunol 2007). Thus, TLR and cytokine-derived signaling in macrophages might be involved in initial down-regulation of PPARg expression and activity after zymosan treatment. We have added this subject in the Discussion (lines 348-352).
(C3) STAT6 phosphorylation was increased 24 hours after zymosan treatment (Fig. 2). How do the authors explain the involvement of STAT6 signaling in regulation of cytokine production in Fig. 3?
(R3) In this model of acute inflammation, increased expression of PPARg and evidence of its activation (ie, CD36, MMR, and Arg1) were shown in WT macrophages and spleen by 24 h following zymosan and were temporally associated with resolution of neutrophilia. Moreover, downregulatory effect of STAT6 deficiency on efferocytic ability of macrophages and an anti-inflammatory proresolving molecule, HGF, proceeds up to 72 h after zymosan treatment. Although macrophages and spleen from STAT6-/- mice also showed up-regulation of PPARg expression and activation, significant delay was demonstrated, and peritoneal inflammation was exaggerated and extended. Please refer to (R5) in response to Reviewer #1’ comment (C5)
(C4) Fig. 4. The authors suggest that STAT6 is involved in PPARg transcriptional activity. However, mRNA expression of PPARg was decreased in STAT6-/- cells. Is PPARg a STAT6 target gene? Are there STAT-binding element(s) in the PPARg promoter?
(R4) Recently, studies from several laboratories support the role of STAT6 in PPARg expression and transcriptional activity. Notably, Liao et al. (J Clin Invest 2011, 121:2736) have shown that transfection of STAT6 induced PPARg promoter-driven luciferase activity in RAW264.7 cells like to the inductive effect of Krüppel-like factor 4 (KLF4) on the PPARg promoter. We have stated this issue in the Discussion as follows (lines 332-335): “Liao et al. [51] showed that transfection of STAT6 induced PPARg promoter-driven luciferase activity in RAW264.7 cells, suggesting a possibility that STAT6 signaling enhances PPARg expression via the inductive effect of STAT6 on the PPARg promoter.”
(C5) Fig. 5. How about the effect of PPARg ligand, such as pioglitazone, on efferocytic ability of peritoneal macrophages from WT and STAT6-/- mice stimulated with zymosan.
(R5) The in vivo experiments from Fernandez-Boyanapalli et al (Blood 2010, 116:4512) demonstrated that in WT mice, pioglitazone treatment did not significantly enhance in vivo macrophage efferocytosis during zymosan-induced peritonitis. However, ex vivo treatment of macrophages of WT mice with pioglitazone (versus vehicle) resulted in enhanced efferocytosis. Notably, they showed that pioglitazone treatment normalized in vivo efferocytosis by chronic granulomatous disease (CGD) macrophages during peritonitis to levels seen in WT macrophages. Macrophage PPARg expression and activity were deficient and acquisition delayed in CGD. Fernandez-Boyanapalli et al. also demonstrated that IL-4/IL13-induced PPARγ signaling enhances efferocytosis specifically (Blood 2009, 113:2047). More recently, data from Cai et. (J Clin Invest Insight 2019, 4:e131355) showed that decreased expression of Arg1 in STAT6-/- microglia/macrophages was responsible for impairments in efferocytosis and loss of antiinflammatory modality. Our previous study demonstrated that in vivo treatment with the STAT6 inhibitor AS1517499 reversed the enhanced PPARg expression and activity induced by apoptotic cell instillation after bleomycin treatment, resulting in inhibition of efferocytosis (Cell Physiol Biochem. 2018;45:1863). Based on these previous findings, we could predict that pioglitazone treatment would enhance in vivo efferocytosis by STAT6 deficient macrophages following zymosan injection to levels seen in WT macrophages.
(C6) There are clear phenotypes of STAT6-/- cells and mice in Figs. 2-5, indicating the involvement of STAT6 signaling. However, there is no evidence showing that STAT6 signaling is involved in PPARg-mediated macrophage function (cytokine production and efferocytosis). Please make the causal relationship clearer.
(R6) We appreciate the reviewer’s suggestion. Data presented here demonstrated that both inflammatory mediators and neutrophilic inflammation were significantly exaggerated and efferocytosis was downregulated in STAT6-/- mice and associated with delayed acquisition of PPARg expression and activation in macrophages. Nonetheless, direct and indirect mechanisms for STAT6-mediated PPARg facilitation in macrophages should be further studied in the context of in vitro and in vivo treatment with zymosan. We have added this issue in the Conclusion (lines 401-403).
We very much appreciate the suggestions raised by the reviewers. We hope that our manuscript will now be accepted for publication in Cells.
Sincerely,
Jihee Lee Kang
Professor
Department of Physiology School of Medicine
Ewha Womans University
Round 2
Reviewer 3 Report
All comments have been addressed with additional experimental data and discussion.
Author Response
According to the Reviewer’s suggestion, we have double checked English spelling. Finally, we have corrected from ‘approache’ to ‘approach’ and from ‘Annexin A1’ to ‘annexin A1’.